# Comparing Hospital Efficiency: An Illustrative Study of Knee and Hip Replacement Surgeries in Spain

**DOI:** 10.3390/ijerph20043389

**Published:** 2023-02-15

**Authors:** Sophie Gorgemans, Micaela Comendeiro-Maaløe, Manuel Ridao-López, Enrique Bernal-Delgado

**Affiliations:** 1Department of Business Management and Organization, School of Engineering and Architecture, University of Zaragoza, 50009 Zaragoza, Spain; 2Institute for Health Sciences in Aragon (IACS), 50009 Zaragoza, Spain

**Keywords:** Malmquist index, data envelopment analysis, knee replacement, hip replacement, in-hospital mortality ratio, public hospitals

## Abstract

WHO’s Health Systems Performance Assessment framework suggests monitoring a set of dimensions. This study aims to jointly assess productivity and quality using a treatment-based approach, specifically analyzing knee and hip replacement, two prevalent surgical procedures performed with consolidated technology and run in most acute-care hospitals. Focusing on the analysis of these procedures sets out a novel approach providing clues for hospital management improvements, covering an existing gap in the literature. The Malmquist index under the metafrontier context was used to estimate the productivity in both procedures and its decomposition in terms of efficiency, technical and quality change. A multilevel logistic regression was specified to obtain the in-hospital mortality as a quality factor. All Spanish public acute-care hospitals were classified according to their average severity attended, dividing them into three groups. Our study revealed a decrease in productivity mainly due to a decrease in the technological change. Quality change remained constant during the period with highest variations observed between one period to the next according to the hospital classification. The improvement in the technological gap between different levels was due to an improvement in quality. These results provide new insights of operational efficiency after incorporating the quality dimension, specifically highlighting a decreasing operational performance, confirming that the technological heterogeneity is a critical question when measuring hospital performance.

## 1. Introduction

Improving healthcare quality while seeking better financial equilibrium has been one of the main priorities of healthcare strategies in the last decades in Europe. Older population structures that require a different healthcare perspective facing the management of chronicity and the incorporation and use of more costly health technologies are major challenges. This prompts healthcare providers to be more concerned with policies that enhance resource productivity [1,2] and quality [3,4,5,6,7,8]. In this regard, Health Systems Performance Assessment (HSPA) is an essential tool to benchmark healthcare providers, especially in the context of the Spanish National Health System (SNHS) where risk transfer is not present and hospital activity is purchased on a retrospective lump-sum basis with no incentive to improve value in health [9]. HSPA framework suggests monitoring a set of dimensions such as equity in access, sufficiency in terms of financial endowment, and quality as the central focus, considering the latter as a nested matrix of the sub-dimensions: patient-centered care, efficiency, and patient safety [10]. Additionally, the performance assessment could be focused on the service provider or on a patient-centered care orientation, which corresponds to the recent treatment approach.

However, HSPA criteria are measured as independent events, whereas many trade-offs are possible. For example, quality and efficiency—usually reported separately—are both closely related when aiming for the maximization of healthcare value (i.e., increasing efficiency while improving quality) [9]. Identifying the impact of the quality of healthcare services on efficiency is a hot topic in health economics and healthcare systems management [11]. In fact, many researchers argued that most of the productivity growth in healthcare has come in the form of quality improvement rather than cost containment [12]. 

Another dimension that might be considered in hospital productivity assessment is the classification of hospitals on the basis of diseases control priorities [13], leading to three different level of hospitals (primary, secondary, and tertiary level) in order to study the effect of possible differences in the scope of services management. Taking into account that hospital outcomes may vary across types of patients, although differences in age and morbidity are captured by adjusting the expected outcome, it could be helpful to stratify hospitals according to their treated complexity [9]. In this sense, classifying hospitals using an approximation of their handled complexity when analyzing a specific surgical procedure allows the performance assessment to work with a better patient homogeneity.

Focusing on the acute-care hospital sector, the aim of this study is to jointly assess technical efficiency and quality using a treatment-based approach, matching the great thrust toward redefining the health sector’s output as disease treatments, rather than as medical goods and services. An advantage of the disease-based approach is that it captures shifts between providers. In theory, a treatment-based approach aggregates all resources for a particular disease—regardless of where or by whom they are provided. This paper analyzes total knee or total hip replacement (TKR or THR), two prevalent procedures with consolidated technology which are performed in most hospitals.

## 2. Materials and Methods

### 2.1. Design and Population

This was an observational study based on administrative data records of discharges from all public acute-care hospital admissions in Spain between 2010 and 2018. Admissions for TKR or THR surgical procedures in those older than 40 years were selected. Hospitals not practicing these surgical procedures during the whole period or performing less than 30 per year were left out to prevent statistical noise. Finally, 220 hospitals were analyzed, accounting for 79% of all the SNHS acute-care hospitals, and for the 97.3% of all the TKR or THR surgical procedures performed during the period of analysis. 

To ensure comparability, SNHS acute-care hospitals were stratified into three groups according to the average severity of the episodes attended at each hospital. For this purpose, the variable containing the hospital *All Patient Refined by Diagnosis Related Groups* (APR-DRG) weights was divided in terciles. Group 1 corresponds to hospitals attending to the least complex episodes (*n* = 37), with an average APR-DRG weight less than 0.7703; group 2 corresponds to hospitals attending to the middle level of complex cases (*n* = 92), with average APR-DRG weight ranging from 0.7704 to 0.8519; and group 3, gathering hospitals attending to the most severe episodes (*n* = 91), corresponds to those hospitals with an average APR-DRG weight above 0.8520.

### 2.2. Sources of Information

Three main data sources were used: (1) The Annual Hospital Survey (SIAE) from which inputs and outputs from the data envelopment analysis (DEA) were retrieved [14]. (2) APR-DRGs weights using the APR-DRGs grouper licensed to the AtlasVPM group by 3M [15]. (3) The micro-data infrastructure on clinical and administrative hospital data, maintained by the AtlasVPM project, which contains information on the date of admission, age and sex of the patient, burden of disease, type of hospitalization, and length of stay, among others variables.

### 2.3. Variables

Two input factors were chosen, the “functioning beds” in the traumatology service as proxy for physical capital (x1) and the “full time equivalent traumatologists” as human capital (x2) and two output factors were selected, “knee and hip replacement episodes” (y1) and “traumatology outpatient visits” (y2). To jointly assess quality as an outcome dimension we used the in-hospital standardized risk-adjusted mortality ratio (SMR) (a1).

### 2.4. Main Endpoint: The Metafrontier Malmquist Productivity Index (MMPI)

The traditional approaches used to estimate efficiency change over time were Windows analysis [16] and the Malmquist index [17]. With the first approach, DEA is applied successively assuming implicitly that there are no substantial technical changes in overlapping time periods of constant width. With the Malmquist index, the assessment of productivity change over time can be carried out assuming technological changes. In the health sector, it can be expected that there is often considerable variation among patients in how inputs are consumed and services are provided, not forgetting that laws and advances in health technology (including medications, procedures, and systems) may also introduce substantial variation in the production possibilities of the hospitals. Under this consideration, this study used a Malmquist productivity index (MPI) under the metafrontier framework, jointly assessing productivity growth and quality changes. This is the first attempt to compare productivity changes between hospitals performing two specific surgeries under different production frontiers.

### 2.5. Analysis

The quality outcome variable or in-hospital standardized risk-adjusted mortality ratio was calculated as the quotient of the observed to the estimated deaths after a TKR or THR surgical procedure. The estimated deaths were obtained through the specification of a logistic multilevel regression model, thus capturing the effect of clustering on estimation (surgical episodes nested into hospitals) [18]. The specified model included as regressors age, sex, and the presence (or absence) of the Elixhauser comorbidity conditions [19,20].

On the other hand, in a nonparametric framework, this study measured productivity growth over time using the Malmquist index, defined as a ratio of distance functions [21], while developing a straightforward computational procedure to calculate the Malmquist index relative to nonparametric frontier technologies which shows that it can be decomposed into technical efficiency changes and technology shifts [21]. The distance functions were estimated using DEA, which are not multiplicatively separable in attributes and inputs/outputs. 

It might be appropriate to measure quality as an output since hospitals must make some trade-offs between the use of resources (quantities or volume and costs) and the quality of care in the current constrained context. Furthermore, the quality-incorporating distance function [21] makes feasible the assessment of the quality change (improvement or deterioration). 

Different types of hospitals may have different technologies and operation scales and may be affected in their capabilities and willingness to quickly respond to market conditions and to adopt technical innovations. We have assumed that technology in operations may vary from one group of hospitals to the other based on the average severity of the episodes attended. Technology heterogeneity is considered a key question in measuring efficiency and productivity of hospitals [22]. Therefore, we have adopted methodology focusing on this heterogeneity.

The Malmquist productivity index (MPI) is defined by the distance function, where x are our inputs, y are our outputs, and a is the quality variable:(1)$$MPI_{t.t+1}^{k}=[\frac{D_{t}^{k}\left(x_{t},y_{t},a_{t}\right)}{D_{t}^{k}\left(x_{t+1},y_{t+1},a_{t+1}\right)}\;{\frac{D_{t+1}^{k}\left(x_{t},y_{t},a_{t}\right)}{D_{t+1}^{k}\left(x_{t+1},y_{t+1},a_{t+1}\right)}]}^{1/2}$$

The related distance functions can be calculated by the linear programming problems, solved t times in which a constant return to scale (CRS) technology is assumed and the model is input oriented. 

The quality change index (QCH) is defined as:(2)$$QCH_{t.t+1}^{k}=[\frac{D_{t}^{k}\left(x_{t},y_{t},a_{t}\right)}{D_{t}^{k}\left(x_{t},y_{t},a_{t+1}\right)}\;{\frac{D_{t+1}^{k}\left(x_{t+1},y_{t+1},a_{t}\right)}{D_{t+1}^{k}\left(x_{t+1},y_{t+1},a_{t+1}\right)}]}^{1/2}$$

Under the constant returns to scale (CRS) assumption, Equation (1) is divided into technical efficiency change (EFFCH) and technological change (TECHCH):MPI = EFFCH × TECHCH(3)

All the indexes collected so far—MPI, QCH, EFFCH, and TECHCH—being > 1 means that, respectively, the productivity, quality, efficiency, and technology change increases over time; otherwise, it indicates that they decrease with time. If these indexes equal 1, it shows that they do not change with time.

Since we assume that the productivity of a hospital operating under one type of technology should not be compared to that of other hospitals operating under different types of technology, we measure the metafrontier MPI (MMPI), as proposed by Chen et al. [23], to highlight the differences in technologies used in different groups of hospitals. The MMPI is defined as:(4)$$MMPI_{t.t+1}^{\;}=[\frac{D_{t}^{*}\left(x_{t},y_{t},a_{t}\right)}{D_{t}^{*}\left(x_{t+1},y_{t+1},a_{t+1}\right)}\;{\frac{D_{t+1}^{*}\left(x_{t},y_{t},a_{t}\right)}{D_{t+1}^{*}\left(x_{t+1},y_{t+1},a_{t+1}\right)}]}^{1/2}$$

“*” denotes the metafrontier. By resolving linear programming problems with the same conditions as before for all hospitals, the related input distance functions can be calculated. The MMPI could be decomposed as the MPI in EFFCH and TECHCH. Moreover, it is possible to obtain a ratio informing about the distance of each hospital to the frontier (for hospital k) and metafrontier (*), i.e., the reciprocal of D*_t_(.)/D^k^_t_ (.) defined as the technology gap ratio (TGR). The ratio (TGRC) describes the change in TGR by giving an idea of the importance of technical differences from one period to the next.

Introducing quality in the model offers an additional index, the quality index (QC) which states the improvement (deterioration) in the technology gap due to a quality change from a_t_ to a_t+1_ (read Chen et al. for in depth details).

The methodology of DEA simultaneously allows for dealing with several inputs and outputs and distinguishes the relative efficiency by means of optimizing the production site. In this way, it is often used to measure the technical effectiveness of healthcare institutions.

Assuming that the hospitals analyzed were using different production technologies, based on patients’ severity differences, showing the adaptability of this model, we applied the Kruskal–Wallis H nonparametric test to look at the technical differences between the different groups, calculating the TGRC over the period or the technical differences between the group boundary and the common boundary.

## 3. Results

As shown in Table 1 of the descriptive statistics of hospitals altogether, there are large differences in the physical and human resources by hospital. There are also significant differences in the quality indicator’s mean. Meanwhile, Table 2, informing about the inputs and outputs by subgroups according to the severity of the episodes attended in the hospital (so called “complexity level”), shows that hospitals dealing with most severe cases exceeded the other two groups: high inputs related to high production levels. Regarding the quality variable, hospitals dealing with the least severe cases have a better performance (mean for a1 = 0.754). Dealing with a higher level of complexity was related with a lower deviation of the in-hospital mortality rate but the median for this group reached 0.79 (CI_95%_: 1.0–1.19) while for both others the median was 0.001 and their measures of the combined weight of a distribution’s tails relative to the center of the distribution (kurtosis) were less significant (9.96 for group 1 and 4.32 for group 2 vs. 5.75 for group 3) so it is not possible to conclude that hospitals dealing with higher severity leads to lower quality over time.

In Table 3, the panels show the productivity changes of hospitals and factors influencing them. The geometric mean of MMPI in Panel A over the analysis period was 0.986, indicating that productivity declined between 2010 and 2018. This result is conditioned overall by the technological deterioration in several periods (TECHCH < 1). The efficiency change presented an upgrading in most of the periods and for the whole period 2010–2018 a status quo with a value close to 1 (EFFCH = 0.997). A detailed analysis of the MMPI indicated that QCH (geometric mean: 0.999) was near 1 over the period, showing a steady trend in spite of declining technological change in hospitals with the lowest severity. Given the decomposition of the quality index, hospitals in this panel reported an improvement in the technological gap as a result of a year-over-year change in quality (QGRC > 1). In addition, the TGRC_10–18_ close to 1 (TGRC = 0.997) shows a slight technical gap between hospitals dealing with the least severe cases and the other two groups. This effect was once again due to the value obtained for the last period while between 2010 and 2016 TGRC was >1 indicating an improvement in terms of changes in the technical gap of this group in comparison with the other two. As a matter of fact, the technology gap ratio (TGR, see Table 4) indicates that the frontier of group 1 is far enough from the metafrontier.

The productivity changes of hospitals of group 2 (hospitals with a medium level of complexity) and factors influencing it are shown in Panel B. The average MMPI among the periods was lower than 1 (MMPI = 0.995) indicating that the productivity slightly decreased. In parallel, however, the MMPI was > 1 with a frequency of 50%. This is caused, as in group 1, by the technological change (TECHCH = 0.997) which was observed <1 in four of the nine paired periods. A quality change >1 was observed in six periods. The technical gap ratio change (TGRC = 0.998) was <1 which shows that TGR change from t to t + 1 was unfavorable (TGR arithmetic mean = 0.768, in Table 4) but less than in the first group. Considering the decomposition of the quality index, the QCH shows a similar behavior than for the first complexity group whereas hospitals of this panel presented a progress in the technology gap due year-over-year change in quality (QGRC > 1).

Panel C shows the productivity of the hospitals in the highest complexity level. The average MMPI of these hospitals during the period was equal to 1 which indicates that the productivity remains constant over time. Over the period, the productivity oscillated, decreasing from 2013 to 2016 but increasing in the rest of the years. In the period 2014–2015, all hospitals had TECHCH > 1, while 69% of them presented weak results in terms of EFFCH (EFFCH < 0.80). Even so, since 2015, the efficiency improved. Their performance in terms of quality gaps (QGRC = 1.369) was lower than in both previous groups due to poor results in QCH in 2010–2011 and 2013–2014. The behavior in QCH was very similar to the previous groups except for in 2010–2011 and 2017–2018. Looking at the decomposition of the MMPI, the factor which contributed to the better performance was the technological change (TECHCH = 1.013), notwithstanding up and down movements, with higher variability than in the first and second complexity levels.

Figure 1 illustrates the changes in productivity (means) and their decomposition over the period 2010–2018. Generally, hospitals dealing with the lower levels of severe episodes of care, groups 1 and 2, showed similar behavior with a year’s delay in the evolution of MMPI during the period (except in 2013), although with a little advantage of group 1 over group 2 (+1.1% vs. −0.3%, respectively). In the third group of hospitals, MMPI showed a relative stability close to 1 until 2015 and during the period remained in status quo. The index indicates that the differences between hospitals with various technologies were increasing year by year. 

Observing the technology gap due to quality change (QGRC) one can see that for all hospitals, independently from the group they belong to, the mean of QGRC shows an improvement in each paired period (value > 1). QGRC was relatively stable in group 1. Hospitals of groups 1 and 2 presented the same behaviors with a one-year gap until 2015 while hospitals of group 3 showed a relative stability from 2010 to 2013 and a high loss in the technology gap in 2014, followed by major improvements in the last three periods. The convergence of all groups of hospitals in the period 2015–2016 is particularly interesting. 

Finally, the technology gap ratio (TGRC) presented higher variation in the first group compared to both others, confirming that the gap comparing other technologies is more important. This group of hospitals showed a wide variability in all decompositions over the periods. On the contrary, while MMPI in group 1 improved first and fluctuated then with up and down movements, as to the decomposition factor, TGRC trended similarly and rapidly rose and decreased after reaching a maximum in 2016.

TGRC and QGRC fluctuated in the same direction (both > 1) mainly in the group of hospitals attending the low level of severe episodes, indicating that the improvement in the technological gap between different groups was due to an improvement in quality (every year except in 2011 and 2017). The technical evolution was associated with improvements in the in-hospital mortality ratio.

The technology gaps between the different groups of hospitals (Table 4) were as higher as the severity attended was lower, showing a clear deterioration over years. Moreover, 31% of hospitals in the third group obtained a TGR ≥ 1 indicating that these units are the closest to the metafrontier (the distance function for this group is the same as the metafrontier distance function) and the median reached 0.93 when, for both other levels it was 0.69 (first level) and 0.82 (second level). Hospitals attending to the most severe episodes are closer to the metafrontier than others.

According to Kruskal–Wallis test results (H K-S in Table 4), the null hypotheses were rejected along the period. On average, TGRs significantly differed from one group to another during the period (*p*-value < 0.05). This suggests that different technologies prevailed in all three groups of hospitals, justifying the use of the metafrontier approach.

## 4. Discussion

Both surgeries studied in this paper, THR or TKR, are frequent surgical conditions in the population and are provided by all acute-hospitals. According to our data about 62% are treated in hospitals with the highest average severity of the episodes attended (group 3), 30% in group 2, and the rest in hospitals of group 1. Mortality associated with these episodes is very infrequent, although severity of the disease could increase with age, diabetes mellitus, or obesity [24]; these comorbidities are well captured by the risk-adjusted specified model. 

The results demonstrate that there were trends in decreasing operational performance of public hospitals in these procedures. When the productivity change was decomposed, the average changes of most components in each complexity level were close to 1, except in the case of the technological change that indicates differences between the first two groups and the last one. According to the results, the hospitals decreased in technical efficiency from 2010 to 2014, increasing from there on regardless of the severity attended.

There were no factors promoting the dynamical operational efficiency of hospitals in the first group despite the MMPI which was > 1 with a frequency above 47.6%. The poor performance of groups 1 and 2 was detected essentially in the technological change (TECHCH = 0.992 and 0.997, respectively). This suggest that hospitals with low complexity decreased their productivity mostly through technological deteriorations. On the other hand, hospitals with the highest severity of the episodes attended show a constant productivity for the overall period due to the positive contribution in terms of technological change.

Looking at the decompositions of the MMPI we observe that, in general, hospitals with low levels of complexity performed relatively stable in EFFCH and TECHCH regarding the upper level. Indeed, in group 3, for every year, EFFCH and TECHCH moved toward the opposite direction; that is, EFFCH declined (improved) as TECHCH improved (declined), which finally resulted in slight changes in MMPI. For the period 2014–2015, we observed the largest technological change in hospitals of group 3, but this effect was totally absorbed by the deterioration in efficiency. In group 1 in 2011–2012, exceptionally, both changes operated in the same direction leading to a growth of the differences compared to the metafrontier. The condition of TECHCH and EFFCH, as in group 3, might be because internal management and operations in TKR and THR have not been synchronously adjusted when these hospitals faced rapid technological or organizational innovation over time. Thus, hospitals absorbed and applied the innovative technology or management process, leading to a tradeoff relationship between EFFCH and TECHCH. The quality change QCH fluctuated widely in the third group and the more substantial change is observed for all hospitals when comparing all periods.

### 4.1. Comparisons with Other Studies

This study reveals that the decrease in the productivity is mainly due to a decrease in the technological change while previous studies have shown significant improvements in different regions of Spain. In the context of the SNHS, in a study of 230 hospitals for the period 2010–2012, the authors observed a decrease in the technical efficiency of hospitals, partially offset by an improvement in the technological frontier, leading to a global improvement in terms of productivity. Nevertheless, detailing each pair period, they observed fluctuating movements in EFFCH (improvement) and TECHCH (decrease) which are similar to those we obtained for the same pair period [25]. The difference in the length of the period could explain the variation for the overall period. Moreover, that study did not consider efficiency and quality together. 

Other studies also covered a large sample of SNHS hospitals using DEA but none of them used the relationship between efficiency and quality [26,27,28]. Differences in findings can be summarized by various factors. These comprise the distinct period of time, measurement of efficiency with a static method for a single time period, and a different configuration of the input-output matrix, working with restrictions over the hospital size and making their results not comparable with those of our study. In a recent international review [11], 62% of studies incorporated quality directly into the efficiency model and thus assume that quality affects the efficiency feasible frontier. Another 35% are two-stage which take into account quality in a second stage. Most studies incorporated outcome quality; however, it is less common to consider the outcome quality related to a particular process. 

On the other hand, we are using a standard outcome quality indicator in hospital efficiency studies by considering the risk-adjusted in-hospital mortality ratio in THR or TKR. Nonetheless, many factors beyond the control of the provider may impact hospital mortality, including pre-hospital care, patient characteristics, and other environmental factors. In this research, the quality indicator is not invariant for the patient because it is based on risk adjustment to capture health heterogeneities among the patient population. 

Beyond the possible controversy of the results, it should be noted that studies assessing efficiency at a highly aggregated level, such as at the hospital level, cannot include all available process indicators and cannot specify all available risk-adjusted mortality rate as a quality variable. In this sense, our study is interesting because it focuses on selected surgical procedures performed in all acute hospitals in Spain, thus avoiding the problem of dimensionality and ensuring homogeneity in order to be compared with each other.

### 4.2. Implications

This study aimed to provide new insights of operational efficiency with the developed quality incorporated MPI, as based on a metafrontier framework, for a sample of Spanish public hospitals. The common frontier of production is appropriate for investigating the relationship between multiple inputs and outputs within different technology groups and the MMPI estimation approach allowed us to compare productivity changes of hospitals operating under different technologies depending on the hospital complexity level and to evaluate the reasons for changes in productivity, i.e., quality factors, efficiency factors, or technical factors. Presently, there is no further empirical research exploring surgical procedures and hospital performance in our context. In this study, we fill the gap in health sector-specific contributions to the measurement of process efficiency and respond to the growing demand in the health economics literature to consider the quality of health care.

### 4.3. Limitations

The use of DEA-type methods presents some limitations. The main problem to be solved in its application to health organizations are the following: sensitivity of the results to the presence of outliers; need to control for the heterogeneity of decision-making units; how to control for differences in quality of medical care; and adequate definition of resources and products [29]. However, with this study and others, DEA and dynamic efficiency controlled by the Malmquist index has been shown to be a powerful instrument that offers policy makers and managers a valuable tool to find the path to improve performance.

This work is innovative in two senses: (i) applying productivity measurement to very prevalent and specific surgical procedures such as TKR and THR in Spanish acute-care public hospitals, and (ii) evaluating the productivity of hospital providers by analyzing the decomposition of the metafrontier Malmquist productivity index.

In addition, patients with TKR and THR diagnoses are not at a high risk of suffering the consequences of fragmented care, such as a lack of coordination and poor communication between multiple healthcare providers, as could happen in the case of other diseases (e.g., hearth failure). The percentage of deaths in this service is therefore generally very low. The available statistics on the causes of death in this kind of surgery highlighted that most of them were the result of complications during and following surgery that require optimal care at health facilities, mainly severe bleeding, high blood pressure, sepsis, and thromboembolism. These deaths are ‘mostly preventable’ because the necessary medical interventions exist and are well known. Depending on the availability of data, it would be suitable to enlarge quality to other indicators as patient safety indicators and infection control rates which are very important parameters that also reflect the quality of care and patient centeredness. As quantity and quality are goods that compete for resources it may appropriate to assess a possible trade-off between both over a long period [30]. 

As lines of future research, to tackle the problem of considering hospitals as identical with respect to organizational and environmental conditions, it may be of interest to perform a multivariate regression analysis considering regions, teaching or non-teaching status, or the effect of quality. Then, the one-stage approach and two-stage approach [31] may be suitable in order to measure the outcome quality impact over hospital performance. Both approaches may be helpful in explaining differences in efficiency across health care providers and may be relevant for policy makers. Some problems of MPI evaluation may appear with the existence of successive period technologies and can be solved using one technology in a period used as reference [32]. Therefore, instead of measuring productivity with reference to a contemporary frontier, it would be executed with reference to a sequential frontier. Finally, to solve the problem of conceiving quality as a descriptive characteristic of the surgical care and not a function of the hospital the dynamic network DEA may be performed, which foresees quality and surgical care as sub-units of the overall care delivery process in the hospital [33].

## 5. Conclusions

The results demonstrate decreasing operational performance and confirm that technology heterogeneity is a crucial issue in the measurement of the performance of hospitals. Our research points at a new direction for measuring the quality/productivity of two specific surgeries. This methodology is especially valuable for healthcare system regulators as an approach to evaluate healthcare procedures in Spain in the effort to consider the quality of surgeries in hospitals. Another contribution of this study is linked to the fact that it was performed on a national scale and over a long period of time (nine years).

## Figures and Tables

**Figure 1 ijerph-20-03389-f001:**
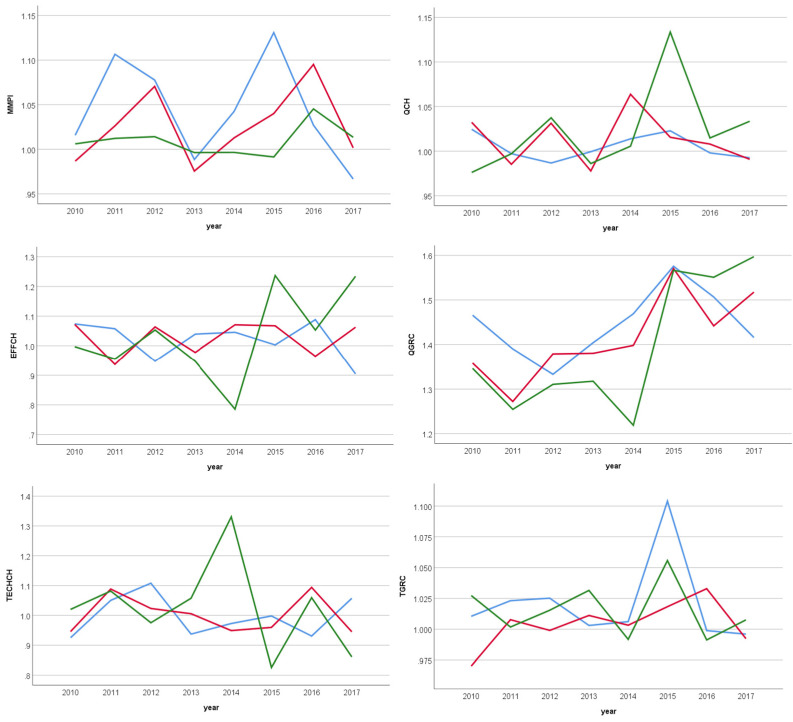
Productivity changes and their decomposition from 2010 to 2018. Note: Blue lines stand for hospitals attending the least severe episodes (group 1). Red lines stand for hospitals attending the middle level of severe episodes (group 2). Green lines stand for hospitals attending the most severe episodes (group 3).

**Table 1 ijerph-20-03389-t001:** Hospital descriptive statistics without classification.

	Median	Mean	CI_95%_	Std. Dev	Min	Max.
Functioning beds (x_1_)	26	32.57	31.7–33.6	23.55	4	133
FTE traumatologists (x_2_)	13.5	16.49	16.0–17.0	11.08	1	67
TKR or THR (y_1_)	225.0	283.38	274.6–292.1	198.25	30	1165
Traumatology Outpatient visits (y_2_)	6687.5	8932.16	8552.4–9311.9	8615.45	68	123,570
In-hospital risk-adjusted mortality ratio (a_1_)	0.001	0.9659	0.894–1.038	1.6379	0.001	13.14

Note: FTE: Full time equivalent; TKR: total knee replacement; THR: total hip replacement.

**Table 2 ijerph-20-03389-t002:** Hospital descriptive statistics by complexity level.

Hospitals	Group 1 (*n* = 37)			Group 2 (*n* = 92)			Group 3 (*n* = 91)		
	Median	Mean	Std. Dev	Median	Mean	Std. Dev	Median	Mean	Std. Dev
Functioning beds (x_1_)	12	11.7	5.73	22	23.11	9.38	44	50.61	25.4
FTE traumatologists (x_2_)	7	6.57	2.39	11	11.64	4.19	22	25.43	11.52
TKR or THR (y_1_)	109	128.43	100.32	192.5	208.24	97.7	369	422.35	215.71
Traumatology Outpatient visits (y_2_)	2420	2625.5	1530.9	5244	5726.5	2,561.4	12,139	14,737.3	10,566.9
In-hospital risk-adjusted mortality (a_1_)	0.001	0.754	2.09	0.001	0.924	1.684	0.791	1.094	1.346
CI (mean)_95%_	0.5–1.0			0.8–1.0			1.0–1.2		

**Table 3 ijerph-20-03389-t003:** Quality index mean incorporating the Malmquist index and its decomposition by type of hospital.

PANEL A Hospitals Group 1	2010–11	2011–12	2012–13	2013–14	2014–15	2015–16	2016–17	2017–18	2010–18 *
MMPI	1.016	1.106	1.078	0.988	1.042	1.131	1.027	0.967	**0.986**
EFFCH	1.074	1.058	0.948	1.04	1.046	1.003	1.088	0.904	**0.997**
TECHCH	0.925	1.05	1.108	0.937	0.973	0.998	0.931	1.057	**0.992**
TGRC	1.01	1.023	1.025	1.003	1.006	1.104	0.999	0.996	**0.997**
QCH	1.024	0.997	0.987	0.999	1.014	1.023	0.998	0.993	**0.999**
QGRC	1.466	1.39	1.333	1.404	1.469	1.575	1.507	1.416	**1.398**
**PANEL B** **Hospitals Group 2**	**2010–11**	**2011–12**	**2012–13**	**2013–14**	**2014–15**	**2015–16**	**2016–17**	**2017–18**	**2010–18 ***
MMPI	0.987	1.026	1.07	0.976	1.013	1.04	1.095	1.002	**0.995**
EFFCH	1.072	0.937	1.064	0.977	1.071	1.068	0.964	1.063	**0.999**
TECHCH	0.946	1.088	1.023	1.005	0.949	0.96	1.094	0.945	**0.997**
TGRC	0.97	1.008	0.999	1.011	1.003	1.018	1.033	0.992	**0.998**
QCH	1.032	0.985	1.031	0.978	1.064	1.016	1.008	0.991	**0.999**
QGRC	1.359	1.272	1.379	1.38	1.398	1.57	1.442	1.517	**1.389**
**PANEL C** **Hospitals Group 3**	**2010–11**	**2011–12**	**2012–13**	**2013–14**	**2014–15**	**2015–16**	**2016–17**	**2017–18**	**2010–18 ***
MMPI	1.027	1.012	1.014	0.996	0.996	0.991	1.045	1.013	**0.999**
EFFCH	0.997	0.955	1.053	0.947	0.786	1.237	1.053	1.243	**0.986**
TECHCH	1.02	1.082	0.975	1.057	1.331	0.826	1.059	0.861	**1.013**
TGRC	1.006	1.002	1.015	1.031	0.992	1.056	0.991	1.008	**1.001**
QCH	0.976	0.997	1.037	0.986	1.006	1.134	1.015	1.033	**0.995**
QGRC	1.347	1.255	1.311	1.318	1.219	1.566	1.551	1.597	**1.369**

* Numbers highlighted in bold are the geometric mean during the periods. EFFCH: technical efficiency change; TECHCH: technological change; TGRC: technology gap ratio change; QCH: quality change; QGRC: quality gap ratio change.

**Table 4 ijerph-20-03389-t004:** Mean difference test for the technology gap ratio (TGR).

	Group 1 *	Group 2 *	Group 3 *	*p*-Value **	H K-S
2010	0.72	0.83	0.891	0.000	38.661
2011	0.726	0.812	0.943	0.000	87.194
2012	0.682	0.799	0.945	0.000	105.91
2013	0.728	0.804	0.928	0.000	85.682
2014	0.663	0.792	0.937	0.000	105.86
2015	0.58	0.642	0.961	0.000	140.03
2016	0.663	0.671	0.911	0.000	86.466
2017	0.573	0.741	0.912	0.000	104.67
2018	0.709	0.817	0.906	0.000	54.891
**2010–18**	**0.672**	**0.768**	**0.926**	**0.000**	**168.67**

* Bold numbers are arithmetic means. ** The *p*-value tested whether a significant difference in the TGR exists between groups by the Kruskal–Wallis test.

## Data Availability

The access to data supporting the manuscript findings is restricted in accordance with the System Level Security Policy of the Unit for Health Services and Policy Research (ARiHSP) at the Institute for Health Sciences in Aragon (IACS) (www.atlasvpm.org/grupo-coordinador/ARiHSPinformationsecuritypolicy) within the framework of the Spanish legal system, in particular, the Law 37/2007 on the Public Sector Information Reuse, the Law 14/2007 on Biomedical Research and the Law 15/1999 on Personal Data Protection. As a consequence, truncated aggregated data may be accessed upon request throughout ARiHSP data sharing agreement, contacting the senior author of the manuscript Enrique Bernal-Delgado (ebernal.iacs@aragon.es) and/or the legal officer Ramón Launa-Garcés (rlaunag.iacs@aragon.es).

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
