# Peer review of "Comparing Hospital Efficiency: An Illustrative Study of Knee and Hip Replacement Surgeries in Spain"

_ijerph, 2023, doi:10.3390/ijerph20043389_

Round 1

Reviewer 1 Report

1. Does it address a specific gap in the field?
2. Detailed discussion of figure 1 required to be explained and quality of figure 1 required to be improved
3. In the results section, critical analysis of results need to be mentioned for future direction; authors can also add 1-2 good future directions
4. What does it add to the subject area compared with other published material?

Author Response

  1. Does it address a sepcific gap in the field?

Thank you for the comment. We add a brief comment respect to the originality of the paper in studying two surgical procedures. The subject area is filling the gap in the sense that it considers all acute-hospitals in a large period for specific surgical interventions. It was never done before in the SNHS context.

  1. Detailed discussion of figure 1 required to be explained and quality required to be improved

We improve the quality of the Figure 1 by pasting it as JPEG.files and modifying the scales of 0Y to gain in visibility. We reorganize explanation and precise variables. Futhermore, we add the following note: Note: Blue lines stand for hospitals attending to the least severe episodes (group 1). Red lines stand for hospitals attending to the mid severe episodes (group 2). Green lines stand for hospi-tals attending the most severe episodes (group 3), avoiding the repetition of legend in each detailed figure of Figure 1.

We add some explanations:

There were no factors promoting the dynamical operational efficiency of hospitals in the first group despite the MMPI which was > 1 with a frequency above 47.6%. The poor performance of groups 1 and 2 was detected essentially in technological change (TECHCH = 0.992 and 0.997 respectively). This suggest that hospitals with low complexi-ty decreased their productivity mostly through technological deteriorations. On the other hand, hospitals with highest severity of the episodes attended show a constant productiv-ity for the overall period due to the positive contribution in terms of technological change.

  1. In the results section, critical analysis of results need to be mentioned for future direction; authors can also add 1-2 good future directions. We reorganize the limitations:

As lines of future research, to tackle the problem of considering that hospitals are identical with respect to organizational and environmental conditions, it may be of inter-est to perform a multivariate regression analysis considering regions, teaching or non-teaching status, or the effect of quality. Then, the one-stage approach and two-stage approach [31] may be suitable in order to measure the outcome quality impact over hos-pital performance. Both approaches may be helpful in explaining differences in efficiency across health care providers and may be relevant for policy makers. Some problems of MPI evaluation may appear with the existence of successive period technologies and can be solved using one technology in a period used as reference [32]. Therefore, instead of measuring productivity with reference to a contemporary frontier, it would be done with reference to a sequential frontier. Finally, to solve the problem of conceiving quality as a descriptive characteristic of the surgical care and not a function of the hospital it may be perform the Dynamic Network DEA which foresees quality and surgical care as sub-units of the overall care delivery process in the hospital [33].

  1. What does it add to the subject area compared with other published material? We detailed reasons why our study is not comparable to others. The new version is as following:

In the context of SNHS, in a study over 230 hospitals for the period 2010-2012, authors observed there has been a decrease in the technical efficiency of hospitals, partially offset by an improvement in the technological frontier leading to a global improvement in terms of productivity. Nevertheless, detailing each pair period, they observed fluctuating movements in EFFCH (improvement) and TECHCH (decrease) which are similar to those we obtained for the same pair-period [25]. The difference in the length of the period could explain the variation for the overall period. Moreover, this study does not consider efficiency and quality together.

Differences in findings can be summarized by various factors. These comprise distinct period of time, measurement of efficiency with a static method for a single time period and different configuration of the input-output matrix, working with restrictions over the hospital size, making their results not comparable with those of our study.

Reviewer 2 Report

This is a technical report on a critical study on the surgical-specific performance of technical efficiency (TE) or productivity with the adjustment of quality outcomes (e.g., mortality rate). Multiwaves of patient care data in multiple hospitals were performed, using a comparative index of TE. The analysis has adequately revealed hospital variations in TE scores.  Although the complexity classification for surgical patients was used, it is unclear how the severity of patients in varying classes of complication is handled in the analysis. 

Following is a list of questions that should be addressed carefully when TE analysis is being performed:

1. Window Analysis: Clarification on the multi-wave analysis is needed. Can time-varying variables as risk adjusters be introduced in the analysis?

2. Severity of Condition(s):  Is there a severity index of the condition that could be introduced as a control variable in the computation of the stochastic frontier analysis?

3. Heterogeneity of Multilevel Analysis of TE: The limitation in selecting a homogenous group of hospitals should be documented.  TE scores may not be properly compared since the size or volume effect of the procedure(s) performed by the study hospitals is adequately handled in DEA.

4. Future Research: More details should be given on how to compute DEA scores when the quality variables are introduced in the computation. The authors could review the work performed by Professor Yasar Ozcan or Healthcare Management Science Review.

In conclusion, this is a fine paper.  If the authors could address the above concerns, it could result in a better paper.

Author Response

  • Window Analysis: Clarification on the multi-wave analysis is needed. Can time-varying variables as risk adjusters be introduced in the analysis?  Thank you for this comment. This method generalises the notion of moving averages to detect efficiency trends of hospitals over time. To measure productivity, the MI is used, which already quantifies the change in technical efficiency assessing whether the hospital moves away from or approaches its corresponding efficiency frontier between two periods. Time is then already include in the analysis.

  • Is there a severity index of the condition that could be introduced as a control variable in the computation of the stochastic frontier analysis? Thank you for this comment. The reviewer should take in account that we are not in a SFA modelling but in DEA. Meanwhile we take into account this remark to make more understandable what represents the complexity level writing in the abstract, the following sentence:

All Spanish public acute-care hospitals were classified according to their average severity attended to, dividing them into three groups

  • Heterogeneity of Multilevel Analysis of TE. Thank you for the comment. All hospitals considered in the analysis are acute-hospitals belonging to SNHS and practicing at least 30 knee or hip total replacements a year. All in all, hospitals present differences in terms of volume, size and severity of episodes atended we already capture in the model. Taking into account your remark, we rewrite the section Design and population:

Observational study based on administrative data records of discharges from all public acute-care hospital admissions in Spain, between 2010 and 2018. Admissions for TKR or THR surgical procedures in elder than 40 years-old were selected. Hospitals non practicing these surgical procedures during the whole period or performing less than 30 per year were excluded to avoid statistical noise. Finally, 220 hospitals were analysed, accounting for the 79% of all the SNHS acute-care hospitals, and for the 97.3% of all the TKR or THR surgical procedures performed during the period of analysis. To ensure comparability, SNHS acute-care hospitals were stratified into three groups according to the average severity of the episodes attended to at each hospital. For this purpose, the variable containing the hospital All Patient Refined by Diagnosis Related Groups (APR-DRG) weights, was divided in terciles. Group 1 corresponds to hospitals attending to the lowest complex episodes (n=37), with an average APR-DRG weight less than 0.7703; group 2 corresponds to hospitals attending to the middle complex cases (n=92), with average APR-DRG weight ranging from 0.7704 to 0.8519; and group 3, gathering hospitals attending to the most severe episodes (n=91), corresponds to those hospitals with an average APR-DRG weight above 0.8520.

  • Future research. Thank you for the comment. We reorganize the limitations and introduce the problem of conceiving quality as a descriptive characteristic enlarging the future lines of research to the Dynamic Network DEA adding a quote to Khushalani and Ozcan, 2017. 

As lines of future research, to tackle the problem of considering that hospitals are identical with respect to organizational and environmental conditions, it may be of interest to perform a multivariate regression analysis considering regions, teaching or non-teaching status, or the effect of quality. Then, the one-stage approach and two-stage approach [31] may be suitable in order to measure the outcome quality impact over hospital performance. Both approaches may be helpful in explaining differences in efficiency across health care providers and may be relevant for policy makers. Some problems of MPI evaluation may appear with the existence of successive period technologies and can be solved using one technology in a period used as reference [32]. Therefore, instead of measuring productivity with reference to a contemporary frontier, it would be done with reference to a sequential frontier. Finally, to solve the problem of conceiving quality as a descriptive characteristic of the surgical care and not a function of the hospital it may be perform the Dynamic Network DEA which foresees quality and surgical care as sub-units of the overall care delivery process in the hospital [33]

Reviewer 3 Report

Thank you for a well written paper on hospital efficiency! I have some minor issues that need to be addressed before the paper is ready to be published.

Make sure that all abbreviations are spelled out when they first appear in the paper (for example APR-DRG on line 73, DEA on line 82, and FTE in table 1). Also, it may be a good idea to spell them out again when they appear in different parts of the paper in order to enhance readability.

As I see it, you have an exploratory purpose since you do not state any hypotheses. Therefore, the verb “asses” may not be the correct one for the aim of the paper.

Please add a reference to the Annual hospital survey.

For the tables: please add a footnote in the tables spelling out all the abbreviations used in the table.

Since you do not have any hypotheses, I would prefer if you only presented results in the results section. Better to leave the analysis and what it means for the discussion. For example, line 229-231 would preferably be found in a discussion.

Please explain why results from [23-24] are not comparable to your findings.

Author Response

Thank you for all comments you gave us.

  1. Make sure that all abbreviations are spelled out when they first appear in the paper (for example APR-DRG on line 73, DEA on line 82, and FTE in table 1). Also, it may be a good idea to spell them out again when they appear in different parts of the paper in order to enhance readability

We take it into account by spelling out all the abreviations when they first appear in the paper. We add footnotes to the tables to enhance readability. Particularly, in table 1:

Note: FTE: Full time equivalent; TKR: total knee replacement; THR: total hip replacement

  1. Add a reference to the Annual hospital survey

We complete the existing reference [14]: Ministerio de Sanidad – Annual Hospital Survey (SIAE) - Portal Estadístico del SNS - Registro de Altas de los Hospitales Generales del Sistema Nacional de Salud. CMBD. Norma Estatal [Internet]. [cited 2022 Nov 9]. Available from: https://www.sanidad.gob.es/estadEstudios/estadisticas/cmbd.htm

  1. Since you do not have any hypotheses, I would prefer if you only presented results in the results section. Better to leave the analysis and what it means for the discussion. For example, line 229-231 would preferably be found in a discussion.

We have moved the analysis to the discussion section as you suggest.

  1. Please explain why results from [23-24] are not comparable to your findings.

These references are now [25-26] and we explain why results from Pérez et al. [25] and López et al. [26] are not comparable:

In the context of SNHS, in a study over 230 hospitals for the period 2010-2012, authors observed there has been a decrease in the technical efficiency of hospitals, partially offset by an improvement in the technological frontier leading to a global improvement in terms of productivity. Nevertheless, detailing each pair period, they observed fluctuating movements in EFFCH (improvement) and TECHCH (decrease) which are similar to those we obtained for the same pair-period [25]. The difference in the length of the period could explain the variation for the overall period. Moreover, this study does not consider efficiency and quality together.

Differences in findings can be summarized by various factors. These comprise distinct period of time, measurement of efficiency with a static method for a single time period and different configuration of the input-output matrix, working with restrictions over the hospital size, making their results not comparable with those of our study.

Reviewer 4 Report

Comparing hospital efficiency: an illustrative study of knee 2 and hip replacement surgeries in Spain is a good area of research with specific improvements.

The English Language needs attention.

Abstract - Information about the sample and sampling technique used is missing.

The significance of the study is missing in the abstract section. Please add 1-2 sentences.

Originality is missing in the abstract section. Please add 1-2 sentences.

In the introduction section, the citation number [10] is missing. The authors have jumped from 9 to 11. See lines No. 41 and 42.

A literature review is deficient.

Please write the sampling technique used in the methodology section

Analysis and results are acceptable.

Discussion is acceptable.

The references section must comply with the journal requirements.

Author Response

  1. Abstract - Information about the sample and sampling technique used is missing.

Thank you for the comment. Indeed, we perform the model over all public acute-hospitals of our National Helath System. We only reject those with less than 30 episods in TKR or THR or hospitals non practicing these surgical procedures during the whole period to avoid statistical noise. We rewrite a sentence of the abstract: .

All Spanish public acute-care hospitals were classified according to their average severity attended to, dividing them into three groups.

  1. The significance of the study is missing in the abstract section.

We add this sentence:

Focusing on the analysis of these procedures sets out a novel approach providing clues for hospital management improvements, covering an existing gap in the literature..

  1. Originality is missing in the abstract section.

In order to respect the lenght permitted, we hope that with the sentence add above it will be sufficient.

  1. In the introduction section, the citation number [10] is missing. The authors have jumped from 9 to 11. See lines No. 41 and 42.

The citation [10] appears in line 37. The mistake comes from the fact that reference 9 is cited twice.

  1. A literature review is deficient.

Thank you for the comment. We improve the comments over the existing literature and add new literature for future lines of research.

  • Klopp GA. The Analysis of the efficiency of productive systems with multiple inputs ans outputs. University of Illinois at Chicago; 1985 [cited 2023 Jan 23]. Available from: https://www.proquest.com/openview/9ca8cb33d17f4203576cac7589bfdf40/1?pq-origsite=gscholar&cbl=18750&diss=y
  • Färe R, Grosskopf S. Theory and Calculation of Productivity Indexes. In: Eichhorn W, editor. Models and Measurement of Welfare and Inequality. Berlin, Heidelberg: Springer; 1994. p. 921–40
  • Khushalani J, Ozcan YA. Are hospitals producing quality care efficiently? An analysis using Dynamic Network Data Envelopment Analysis (DEA). Socioecon Plann Sci. 2017 Dec 1;60:15–23.
  • Please write the sampling technique used in the methodology section

As we explain above, we did not apply any sampling technique as we perform the model over all public acute-hospitals belonging to our National Health System. We only reject those hospitals which are not attending at least 30 episods in TKR or THR or not practicing these surgical procedures during the whole period to avoid statistical noise. We precise in the abstract: .

All Spanish public acute-care hospitals were classified according to their average severity attended to, dividing them into three groups.

And we rewrite the whole section Design and population:

Observational study based on administrative data records of discharges from all public acute-care hospital admissions in Spain, between 2010 and 2018. Admissions for TKR or THR surgical procedures in elder than 40 years-old were selected. Hospitals non practicing these surgical procedures during the whole period or performing less than 30 per year were excluded to avoid statistical noise. Finally, 220 hospitals were analysed, ac-counting for the 79% of all the SNHS acute-care hospitals, and for the 97.3% of all the TKR or THR surgical procedures performed during the period of analysis.

To ensure comparability, SNHS acute-care hospitals were stratified into three groups according to the average severity of the episodes attended to at each hospital. For this purpose, the variable containing the hospital All Patient Refined by Diagnosis Related Groups (APR-DRG) weights, was divided in terciles. Group 1 corresponds to hospitals attending to the lowest complex episodes (n=37), with an average APR-DRG weight less than 0.7703; group 2 corresponds to hospitals attending to the middle complex cases (n=92), with av-erage APR-DRG weight ranging from 0.7704 to 0.8519; and group 3, gathering hospitals attending to the most severe episodes (n=91), corresponds to those hospitals with an av-erage APR-DRG weight above 0.8520.